IRF7-deficient MDCK cell based on CRISPR/Cas9 technology for enhancing influenza virus replication and improving vaccine production

Mayuramart Oraphan 1
Poomipak Witthaya 2
Rattanaburi Somruthai 1
Khongnomnan Kritsada 3
Anuntakarun Songtham 1
Saengchoowong Suthat 3
Chavalit Tanit 3
Chantaravisoot Naphat cnaphat@gmail.com 3 4
Payungporn Sunchai sp.medbiochemcu@gmail.com 1 3
1 Research Unit of Systems Microbiology, Faculty of Medicine, Chulalongkorn University , Bangkok , Thailand
2 Research Affairs, Faculty of Medicine, Chulalongkorn University , Bangkok , Thailand
3 Department of Biochemistry, Faculty of Medicine, Chulalongkorn University , Bangkok , Thailand
4 Center of Excellence in Systems Biology, Faculty of Medicine, Chulalongkorn University , Bangkok , Thailand
Cui Yingjun
Electronic publication date: 2022 Sep 21
Publication date: 2022
Volume: 10
Electronic Location ID: e13989
Received 2022 Apr 22; Accepted 2022 Aug 11
Copyright: ©2022 Mayuramart et al.
Copyright year: 2022
Copyright holder: Mayuramart et al.
License: This is an open access article distributed under the terms of the Creative Commons Attribution License, which permits unrestricted use, distribution, reproduction and adaptation in any medium and for any purpose provided that it is properly attributed. For attribution, the original author(s), title, publication source (PeerJ) and either DOI or URL of the article must be cited.
License URL: https://creativecommons.org/licenses/by/4.0/

Keywords: CRISPR-Cas9, IRF7, MDCK, Influenza, Interferon, Vaccine production

Funding: Research Unit of Systems Microbiology, Faculty of Medicine, Chulalongkorn University National Research Council of Thailand (NRCT) N41A640077 Funding was provided by the Research Unit of Systems Microbiology, Faculty of Medicine, Chulalongkorn University and the National Research Council of Thailand (NRCT) [N41A640077]. There was no additional external funding received for this study. The funders had no role in study design, data collection and analysis, decision to publish, or preparation of the manuscript.

==============================
The influenza virus is a cause of seasonal epidemic disease and enormous economic injury. The best way to control influenza outbreaks is through vaccination. The Madin-Darby canine kidney cell line (MDCK) is currently approved to manufacture influenza vaccines. However, the viral load from cell-based production is limited by host interferons (IFN). Interferon regulating factor 7 (IRF7) is a transcription factor for type-I IFN that plays an important role in regulating the anti-viral mechanism and eliminating viruses. We developed IRF7 knock-out MDCK cells (IRF7−/ − MDCK) using CRISPR/Cas9 technology. The RNA expression levels of IRF7 in the IRF7−/ − MDCK cells were reduced by 94.76% and 95.22% under the uninfected and infected conditions, respectively. Furthermore, the IRF7 protein level was also significantly lower in IRF7−/ − MDCK cells for both uninfected (54.85% reduction) and viral infected conditions (32.27% reduction) compared to WT MDCK. The differential expression analysis of IFN-related genes demonstrated that the IRF7−/ − MDCK cell had a lower interferon response than wildtype MDCK under the influenza-infected condition. Gene ontology revealed down-regulation of the defense response against virus and IFN-gamma production in IRF7−/ − MDCK. The evaluation of influenza viral titers by RT-qPCR and hemagglutination assay (HA) revealed IRF7−/ − MDCK cells had higher viral titers in cell supernatant, including A/pH1N1 (4 to 5-fold) and B/Yamagata (2-fold). Therefore, the IRF7−/ − MDCK cells could be applied to cell-based influenza vaccine production with higher capacity and efficiency.

Introduction

Influenza viruses are important pathogens that cause respiratory tract infections and numerous public health problems. The influenza virus is a member of the Orthomyxoviridae family. It comprises eight single-stranded RNA segments in a genome that encode 9–12 proteins. The four types of influenza, influenza A, B, C and D, are characterized by nucleoprotein (Puthavathana et al., 2005) and matrix proteins (M). Influenza A and B can cause severe human respiratory disease (Taubenberger & Morens, 2008; ArbeitskreisBlut, 2009; More et al., 2018). There are four main subtypes of influenza circulating in the human population, including influenza A pandemic H1N1 (pH1N1), influenza A H3N2, influenza B Yamagata lineage, and influenza B Victoria lineage (Bedford et al., 2015). Although the influenza A virus is frequently found, the infection rate of the influenza B virus has been increasing, especially in the 2017–2018 influenza season, which witnessed a global outbreak of influenza B. The infection rate of influenza B in the 2017–2018 influenza season was around 30–66% of influenza-positive samples (Blanton et al., 2017; Adlhoch et al., 2018). During such outbreaks, the most important step to prevent influenza is to use the influenza vaccine (Mei et al., 2013; Adlhoch et al., 2018; CDC, 2018). Influenza vaccine has two main types, inactivated vaccine and live-attenuated vaccine. There are two main ways to produce an influenza vaccine. The main one is the embryonated eggs-based process which produces a high viral titer of influenza. Still, this method also has various disadvantages, including the influenza strains being limited to the virus that can be grown in poultry. It is also time-consuming, yields a low production capacity, and has a link to egg component allergies, making this egg-based process obsolete (Manini et al., 2017). A cell-based process substituted the egg-based process with both higher production capacity and a high viral titer (Petiot et al., 2018). Recently, a commercial cell-based influenza vaccine has been made available. The MDCK cell line, approved by the Food and Drug Administration (FDA) and European Medicines Agency (EMA), has been used to produce cell-based influenza vaccines. However, the innate immune response in mammalian cells will restrict the viral propagation, yielding a lower viral titer than the embryonated egg-based vaccine production (Tree et al., 2001).

Type I interferons (IFN) are cytokines which play important roles in inflammation and immunoregulation that responds to viral infection. When the signal transduction of type I interferon begins, the transcription factor of IFN plays an important role in expressing the IFN gene. Interferon regulatory factor 7 (IRF7) is a member of the IRF family that is produced in small amounts as an inactive form in the cytoplasm (Honda, Takaoka & Taniguchi, 2006; Hale, Albrecht & Garcia-Sastre, 2010). IRF7 can form homodimers or heterodimers with other IRFs, resulting in triggering the strong induction of type-I IFN transcription and signal transductions (Honda, Takaoka & Taniguchi, 2006).

According to a previous study, shRNA targeting the IRF7 gene demonstrated a higher viral titer of influenza A (H1N1 & pH1N1) and influenza B (Yamagata lineage) than the control MDCK (Hamamoto et al., 2013). However, the shRNA may affect the gene silencing temporarily (knock-down) by interference with the transcription and translation pathways. Thus, it may be difficult to maintain shRNA in the cell culture for vaccine production. Recently, a bacterial adaptive immune system based on the Clustered Regularly Interspaced Short Palindromic Repeats and CRISPR-associated protein (CRISPR-Cas) has been applied to powerful genome editing machinery with a highly specific ability to knock-out the target gene effectively (Ishino, Krupovic & Forterre, 2018).

The CRISPR-Cas system is activated when foreign nucleic acids or bacteriophages invade a bacterial cell. A part of the viral genome is integrated into interspaced repeats of the CRISPR array, then transcribed to precursor CRISPR RNA (pre-crRNA), and then processed to mature crRNA. Mature crRNA will assemble with the Cas protein and target the viral genome, which is subsequently cleaved. Recently, CRISPR-Cas has been applied to genome editing technology, and CRISPR-Cas9, CRISPR-Cas type II Class 2, plays an important role in investigating cellular biological pathways (Cho et al., 2013; Ma, Zhang & Huang, 2014; Xiao-Jie et al., 2015). Therefore, the IRF7−/− MDCK cell line was developed based on CRISPR-Cas9 technology to provide higher viral titers for influenza vaccine production.

Material and Methods

Single guide RNA (sgRNA) preparation for CRISPR/Cas 9 vector

The single guide RNA (sgRNA) of canine IRF7 was designed from the MultiTargeter web-based tool (https://multicrispr.net/) by using the IRF7 nucleotide sequence (accession number: XM_005631711) (Prykhozhij et al., 2015). After prediction, candidate sgRNAs were selected, as summarized in Table 1. The DNA template for the sgRNA was inserted into a GeneArt® CRISPR Nuclease Vector (Invitrogen, Waltham, MA, USA) and transformed to the E. coli competent cell strain JM109 (RBC bioscience, New Taipai City, Taiwan) following the manufacturer’s instructions. After colony selection by antibiotics, recombinant plasmid DNAs (sgRNA vector) were extracted using the QIAGEN Plasmid Midi Kit (QIAGEN, Hilden, Germany) and then confirmed by Sanger sequencing (Macrogen, Seoul, SouthKorea).

Table 1 Primer and oligonucleotide.

Experiment	Name	Sequence	References	
sgRNA plasmid
construction	IRF7_F2066_TS	TATACCATCTACCTGGGCTTGTTTT	this study	
IRF7_F2066_BS	AAGCCCAGGTAGATGGTATACGGTG	this study	
gDNA amplification	IRF7_F3572	GAACCAGGACACCCCCATCTT	this study	
IRF7_R4032	GGAAGTGTTCCAGGTCCTCGT	this study	
Whole genome sequencing of influenza B virus	FluB cocktail		Modified from Zhou et al., 2014	
Flu B-PBs-UniF	GGGGGGAGCAGAAGCGGAGC	
Flu B-PBs-UniR	CCGGGTTATTAGTAGAAACACGAGC	
Flu B-PA-UniF	GGGGGGAGCAGAAGCGGTGC	
Flu B-PA-UniR	CCGGGTTATTAGTAGAAACACGTGC	
Flu B-HANA-UniF	GGGGGGAGCAGAAGCAGAGC	
Flu B-HANA-UniR	CCGGGTTATTAGTAGTAACAAGAGC	
Flu B-NP-UniF	GGGGGGAGCAGAAGCACAGC	
Flu B-NP-UniR	CCGGGTTATTAGTAGAAACAACAGC	
Flu B-M-Uni3F	GGGGGGAGCAGAAGCASGCACTT	
Flu B-NS-Uni3R	CCGGGTTATTAGTAGTAACAAGAGGATT	
Reverse transcription	Uni_Flu cDNA	IAGCARAAGC	Zhao et al., 2016	
RT-qPCR	IRF7_F3572	GAACCAGGACACCCCCATCTT	this study	
IRF7-mRNA_R1272	CCGTGGCTCCAGCTTCACC	
FluA_M_F151	CATGGARTGGCTAAAGACAAGACC	Suwannakarn et al., 2008	
FluA_M_R276	AGGGCATTYTGGACAAAKCGTCTA	
FluB_PB1_269	AGGCTTTGGATAGAATGGATGA	Saengchoowong et al., 2019	
FluB_PB1_385	AAGTCTGTCTCCCCTGGGTT	
GAPDH-F85	GTGAAGGTCGGAGTCAACGG	
GAPDH-R191	TCAATGAAGGGGTCATTGATGG	

Generation of IRF7 knock-out (IRF7−/−) MDCK cells

Wildtype (WT) MDCK (ATCC® CCL-34™) cells were grown in minimal essential media (MEM) with 10% fetal bovine serum (Hyclone™, GE Healthcare, Chicago, IL, USA) and 1% penicillin-streptomycin (Hyclone™, GE Healthcare, Chicago, IL, USA). To introduce the CRISPR Nuclease OFP Reporter Vector, MDCK cells were plated on a 60-mm cell culture disc in MEM medium without antibiotics and were grown until reaching 70% confluency. Cells were transiently transfected with 5 ng of sgRNA vector and 40 µL of Lipofectamine®2000 (Invitrogen, Waltham, MA, USA) according to the manufacturer’s protocol. After transfection, cells were selected by orange fluorescent protein (OFP) reporter marker in the plasmid using BD FACS Aria™ III cell sorter (BD Biosciences, USA). The IRF7−/− MDCK polyclonal cell line was generated from sorted cells. They were collected and recovered for further experiments (Fig. S1).

DNA extraction and PCR

Genomic DNA of WT MDCK and IRF7−/− MDCK were extracted using the Genomic DNA Extraction kit (RBC bioscience, Taiwan), following the instruction protocol. Genomic DNA of both cell lines was used as templates to amplify the upstream and downstream sgRNA flanking regions within the IRF7 gene by using primers, as shown in Table 1. The PCR mixture consisted of 0.2 µM of each primer, 1X reaction buffer, 200 µM dNTPs, 1.5 mM MgCl2 and 0.5 U Taq DNA polymerase (Biotechrabbit™, Berlin, Germany) in a total volume of 25 µL. PCR reactions were carried out on the GSX1 Master cycler (Eppendorf, Hamburg, Germany) by using the following thermal profile; pre-denaturation at 94 °C for 3 min, amplification for 40 cycles (94 °C for 30 s, 60 °C for 30 s, 72 °C for 45 s) and final-extension at 72 °C for 7 min. After that, the expected PCR product (approximately 460 bp) was purified from agarose gel using the QIAquick Gel Extraction Kit (QIAGEN, Hilden, Germany) and then confirmed by Sanger sequencing (Macrogen, Seoul, South Korea).

Cleavage detection

A cleavage assay was performed using the GeneArt® Genomic Cleavage Detection Kit (Invitrogen, Waltham, MA, USA), following the instruction protocol to detect the locus-specific cleavage of genomic DNA. Briefly, the 106 cells of WT and IRF7−/− MDCK were lysed by lysis buffer and heating (68 °C for 15 min and 95 °C for 10 min). Then the sgRNA targeted region was amplified by PCR as described above. PCR products were denatured (95 °C for 5 min), followed by a rapid temperature decrease of 2 °C/sec from 95 °C to 85 °C, and then a slow temperature decline of 0.1 °C/sec from 85 °C to 25 °C for re-annealing of the PCR product. At this step, the heteroduplex of PCR products was cleaved by Detection Enzyme for 1 h before visualizing on 2% agarose gel. Cleavage efficiency was calculated based on the band intensity of cleavage and parental product by using the following formula: Cleavage Efficiency=1−1−fraction cleaved12,

where the fraction cleaved = sum of cleaved band intensities/(sum of the cleaved and parental band intensities), following the manufacturer’s recommendation.

Relative expression of IRF7 gene by RT-qPCR

The total RNA was extracted from infected (12 h post-infection; hpi) and uninfected WT or IRF7−/− MDCK cells using GenUP™ Total RNA kit (Biotechrabbit™, Berlin, Germany). Then RNA was reverse transcribed to cDNA using the RevertAid First Strand cDNA Synthesis Kit (Thermo FisherScientific, Waltham, MA, USA). Briefly, 40 ng of total RNA from each sample was mixed with 0.2 µg oligo(dT)18 and incubated at 65 °C for 5 min, then chilled on ice for 2 min, followed by the addition of RT mixture containing 1x reaction buffer, 1 mM dNTPs, 20 U RiboLock RNase inhibitor and 200 U RevertAid RT. The RT reaction was incubated at 37 °C for 90 min and was followed by heat inactivation at 70 °C for 10 min. The cDNAs were used as the template for the qPCR reaction of IRF7 and internal control (GAPDH) genes. The Quantitative PCR reaction consisted of 1X Luna® Universal qPCR Master Mix (New England Biolabs, Ipswich, MA, USA), 0.25 µM of each primer and 1 µL of cDNA template in a total volume of 10 µL. The reaction was performed on the StepOnePlus™ Real-Time PCR Instruments (Applied Biosystems, Waltham, MA, USA) by using the following profile: pre-denaturation at 95 °C for 1 min, amplification for 40 cycles (95 °C for 15 s, 60 °C for 20 s and fluorescent detection at 80 °C for 10 s). Quantitation of GAPDH was performed as described previously (Saengchoowong et al., 2019). The relative expression of IRF7 normalized with GAPDH was determined by RT-qPCR using the ΔΔCt method.

Western blotting analysis

Wildtype and IRF7−/− MDCK cells were seeded for 7 ×105 cells to T25 flasks until reaching 80% confluency and then infected with B/Yamagata (B/Massachusetts/2/2012; ATCC®VR-1813™) at MOI of 0.01 as triplicates in each group. Cultured MDCK cells were lysed with a lysis buffer containing 1% Triton X-100, 20 mM Tris HCl (pH 7.4), 150 mM NaCl, 1 mM EDTA, and EDTA-free protease inhibitor cocktail (PIC) (Roche, Basel, Switzerland). Cells were centrifuged at 16,000 x g for 10 min before collecting the supernatant. Total protein concentrations were determined by the BCA protein assay kit (Thermo Fisher Scientific, Waltham, MA, USA). Whole cell lysate (25 µg) from each condition was separated by SDS-PAGE and transferred to a nitrocellulose membrane (Bio-Rad, Hercules, CA, USA). The membrane was blocked in Odyssey® Blocking Buffer (TBS) (LI-COR, USA), probed with primary antibody against IRF7 (Santa Cruz Technology, Dallas, TX, USA) and GAPDH (Cell Signaling Technologies, Danvers, MA, USA) overnight at 4 °C, and incubated with IRDye® secondary antibodies (LI-COR, Lincoln, NE, USA). Finally, the membrane was scanned on the Odyssey® CLx Imaging Systems (LI-COR, Lincoln, NE, USA). The quantitation for western blotting results was performed using ImageJ software.

Sample preparation for RNA-Seq

MDCK cells (WT or IRF7−/−) were seeded for 7 ×105 cells to T25-flask. Cells were incubated for 14-16 h to reach 80% confluent and infected with B/Massachusetts/2/2012 (ATCC®VR-1813™) at a multiplicity of infection (MOI) of 0.01 as described previously (Saengchoowong et al., 2019). To investigate the innate immune response genes during the early phase of viral infection, cells were harvested at 12 hpi for RNA extraction using GenUP™ Total RNA kit (Biotechrabbit™, Hilden, Germany) with on-column DNA digestion per the instruction protocol. At least 3 µg of total RNA was transferred to GenTegra-RNA 0.5 mL screwcap microtubes (GenTegra®, Pleasanton, CA, USA) and put under a vacuum until the tube was dried. Briefly, 1 µg of total RNA with a RIN value above 7 was used for library preparation using the NEBNext® Ultra™ RNA Library Prep Kit for Illumina® (New England Biolabs, Ipswich, MA, USA) according to the manufacturer’s protocol. The library was sequenced in paired-end (150x2) based on the HiSeq 2500 platform (Illumina, San Diego, CA, USA). The NGS service and data analysis were performed by Vishuo Biomedical (Vishuo Biomedical, Technominium, Singapore). The experiment was performed in duplicate.

RNA-Seq data analysis

The raw data (FASTQ) were evaluated with the FastQC software (v.0.11.5). Then sequencing adapters, low-quality bases (Q score <20) and short reads (<50 bp) were removed in the trimming process with Trimmomatic software (v.0.32). The pass-filter reads were aligned to the reference genome of Canis familiaris (HISAT2 index: Ensembl CanFam3.1 genome_tran) using HISAT2 software (v.2.1.0). To identify and quantify the transcripts, the alignment files were analyzed with the Cufflinks software (v.2.2.1). The expression level of each transcript was calculated as fragments per kilobase per million reads (FPKM) using the Cuffdiff software. Differential expressions of IFN-related genes were visualized as a heatmap generated using Heatmapper (http://www.heatmapper.ca/). The volcano plot showing the log2(fold-change) of gene expression profiles in IRF7−/− compared to WT MDCK cells after 12 hpi was generated using GraphPad Prism software (v.9.0). Enrichment analysis of up-regulated and down-regulated gene ontology biological processes (GOBP) was plotted using the DAVID Bioinformatics Resources tools (v.6.8). The −log10 p- value of each GO term and the fold enrichment value were plotted.

Influenza whole genome sequencing

To confirm that the genome of the virus propagated from IRF7−/− MDCK did not differ from the WT MDCK, the next generation sequencing (NGS) was applied for whole-genome characterization. The WT and IRF7−/− MDCK cells were infected with B/Yamagata (B/Massachusetts/2/2012; ATCC®VR-1813™) at MOI of 0.01, then the supernatant was harvested at 48 hpi, which is a common incubation period for influenza viral propagation. According to the instruction protocol, viral RNA was extracted from 200 µl supernatant using the QIAamp® Viral RNA Mini kit (QIAGEN, Hilden, Germany). Then, the cDNA was generated by using RevertAid First Strand cDNA Synthesis Kit (Thermo Fisher Scientific, Waltham, MA, USA) from 10.5 µL of viral RNA, and Uni_Flu cDNA was used for cDNA synthesis (Table 1) (Zhao et al., 2016). The reverse transcription was performed as described above. The 8 segmented genes were amplified using modified primers (Table 1) and repeating the conditions from the previous report (Zhou et al., 2014). The reaction consisted of 1x Phusion HF buffer, 0.35 mM dNTPs, 0.5 µM FluB cocktail primers and 0.5 U Phusion High-Fidelity DNA Polymerase (Thermo Fisher Scientific, Waltham, MA, USA). The PCR products were purified using the QIAquick PCR purification kit (Qiagen, Hilden, Germany) and then fragmented by M220 Covaris® Focused-ultrasonicators with 20% duty factor, 50 units of peak incident power (W), 200 cycles per burst for 150 s (Covaris, Woburn, MA, USA). The fragmented DNA (approximately 200 bp) was used for library preparation by NEBNext® Ultra™ DNA Library Prep Kit for Illumina® (New England Biolabs, Ipswich, MA, USA), followed by paired-end (150 × 2) sequencing on the MiSeq platform (Illumina, San Diego, USA). The raw FASTQ data was trimmed (<Q30) and assembled by CLC Genomics workbench (Qiagen, Hilden, Germany). The genome sequences were compared between influenza B virus obtained from IRF7−/− and WT MDCK.

Hemagglutination (HA) assay

Influenza viral titers in supernatant were determined by HA assay. As a working solution, the turkey red blood cells (TRBC) were washed and resuspended in PBS at a final concentration of 0.5%. The 100 µl of supernatants were 2-fold serially diluted with PBS in 96-well plates with a V-shape bottom, and then 50 µl of 0.5% TRBC suspension was added into each well and incubated at room temperature for 30 min. The HA titers were measured as the highest dilution of the sample showing complete lattice formation.

Absolute quantitation of influenza viral genes

Influenza viruses used in this experiment included A/pH1N1 (A/Thailand/104/2009; accession no. GQ169381 –GQ169385, GQ205443, GQ259597, GQ229379) and B/Yamagata (B/Massachusetts/2/2012; ATCC®VR-1813™). The WT and IRF7−/− MDCK cells were infected with each strain of influenza virus as described above. The RNA extraction and cDNA synthesis procedures were similar to those used for influenza whole genome sequencing. Influenza A (M gene) and influenza B (PB1 gene) were quantified based on qPCR, using specific primers, as shown in Table 1, and with a reaction mixture as described earlier. The StepOnePlus™ Real-Time PCR Instrument (Applied Biosystems, Waltham, MA, USA) was applied for qPCR with the thermal profiles for influenza A (M1 gene) and influenza B (PB1 gene) according to previous studies (Suwannakarn et al., 2008; Saengchoowong et al., 2019). The 10-fold serial dilutions of standard RNA (ranging from 106 to 10 copies/µL) were used as qPCR templates. Then Ct values were plotted against concentrations to construct the standard curve for absolute quantitation.

Statistical analysis

The result was shown as mean and SEM using three technical replicates. The qPCR data were analyzed and visualized by GraphPad Prism v.9.0. The unpaired t-test was used to calculate the significant difference between group expression of IRF7 gene and viral genes quantitation. Statistically significant was determined when the p-value <0.05.

Results

Gene modification efficiency of IRF7−/− MDCK cells

To investigate whether CRISPR-Cas9 effectively affected cleavage of the IRF7 region, the genomic DNA from WT and IRF7−/− MDCK were extracted, and then the sgRNA flanking region was amplified. The agarose gel electrophoresis presented the same length of PCR product (461 bp) in both WT and IRF7−/− MDCK. Despite that, a two bp deletion within the cleavage site of Cas9 was observed from the sequence alignment of PCR products (Fig. 1A). Moreover, the mixed signals downstream of the cleavage site were observed in the sequencing chromatogram of IRF7−/− MDCK, indicating the heterogeneity of the cells after the genome editing process (Fig. 1B). Due to heterogeneity of the cells, the cleavage assay was performed to determine the gene modification efficiency. Parental and cleavage band intensities were applied to calculate cleavage efficiency and gene modification efficiency (Fig. 1C). The gene modification efficiency of the IRF7−/− MDCK cell was approximately 60%.

Figure 1 DNA sequencing and cleavage detection of IRF7−/− and WT MDCK cells from sgRNA flanking region.

(A) The pairwise sequences alignment shows 2 bp deletions within cleavage site of IRF7−/− MDCK cells. The sgRNA and PAM sequences were labeled (B) Sequencing chromatogram shows the heterogeneity of sequence after the cleavage site (red block). (C) The agarose gel electrophoresis of cleavage detection displayed the parental band (black arrow) and cleavage bands (red arrow) from IRF7−/− MDCK. The cleavage efficiency and gene modification efficiency were calculated from cleavage band intensities of samples compared with positive control of cleavage detection (P). (+); with Detection Enzyme, (-); without Detection Enzyme, (N); negative control of cleavage detection.

IRF7 gene expression level between WT and IRF7−/− MDCK cells

To determine the IRF7 gene expression of WT and IRF7 −/− MDCK while being either uninfected or infected with influenza virus, RT-qPCR and western blot analysis was performed. Influenza B virus (B/Massachusetts/2/2012) was used to represent the influenza-infected condition. The relative quantitation was determined from the expression level of IRF7, normalized with the GAPDH gene. The expression of the IRF7 gene was significantly lower in IRF7−/− MDCK cells compared with WT MDCK for both uninfected (94.76% reduction; p = 6.65e−07) and viral infected conditions (95.22% reduction; p = 3.97e−04) as shown in Fig. 2A. Moreover, the expression of IRF7 protein was also significantly lower in IRF7−/− MDCK cells for both uninfected (54.85% reduction; p = 3.40e−03) and viral infected conditions (32.27% reduction; p = 3.10e−03) compared to WT MDCK as shown in Fig. 2B. The significant decrease of IRF7 expression in IRF7−/− MDCK implies that the IRF7 gene in IRF7−/− MDCK cells were knocked out.

Figure 2 Expression of IRF7 and IFN-related genes.

(A) The relative expression of IRF7 gene determined by RT-qPCR and (B) protein expression based on western blot in IRF7−/− MDCK compared with WT MDCK uninfected cells during uninfected condition and infected with influenza B virus (12 hpi). (C) Heatmap representing the average fold-changes of differentially expressed IFN-related genes among different conditions. (*); p-value ≤ 0.05, (***); p-value ≤ 0.001.

RNA expressions of IFN-related genes

To determine the consequences of the IRF7 gene knock-out, high-throughput RNA sequencing was performed in WT and IRF7−/− MDCK while either uninfected or infected with influenza virus (12 hpi). The total RNA extracted from each sample passed the sample quality control with a 9.0–10.0 RNA integrity number (RIN). After quality (≥ Q20) and adapter trimmings, approximately 48-60 million reads (≥ 98% of total reads) were obtained. The differential expression of IFN-related genes between IRF7−/− andWT MDCK in the viral infected condition demonstrated that ISG15, IFNK, MX1, IRF8, IRF6, RSAD2 and OAS2 were down-regulated, whereas IFIT1 was up-regulated in IRF7−/− MDCK cells (Fig. 2C). The result indicated that the interferon-based anti-viral response was significantly deficient in IRF7−/− MDCK cells.

Expression profiling and gene ontology of IRF7−/−MDCK after viral infection

To differentiate the gene expression profiling and gene ontology after viral infection between WT and IRF7−/−MDCK cells, log2 fold changes of the transcripts obtained from the RNA-Seq of IRF7−/−compared to WT MDCK were analyzed. There were 720 dysregulated genes between WT and IRF7−/− MDCK cells infected with the influenza virus. The differential expression included 344 up-regulated genes and 376 down-regulated genes in IRF7−/− compared to WT MDCK cells infected with the influenza virus. The results revealed that when being infected with influenza B virus, various type 1 IFN-stimulated genes responsible for the innate immune response against viruses were significantly down-regulated in IRF7−/− cells, as indicated in orange circles (Fig. 3A). Moreover, the gene ontology biological processes (GOBP) of significantly up-regulated and down-regulated genes in IRF7−/− MDCK cells are shown in Fig. 3B. The genes supporting the defense response to the virus, the negative regulation of viral genome replication and the regulation of interferon-gamma production were decreased in the knocked-out cells. In contrast, up-regulated GO terms included positive regulation of defense response to virus by the host, innate immune response, and I-kappaB-kinase/NF-kappaB signaling (Fig. 3B). From these findings, it could be inferred that, even though there would be some responses from the host cells after viral infection, the virus would be able to propagate more efficiently in this IRF7−/− than in wildtype MDCK cells.

Figure 3 Effects of viral infection on the gene expression profiling and gene ontology of IRF7−/− compared to WT MDCK cells.

(A) Volcano plot showing the expression profiling comparison between IRF7−/− and WT MDCK cells after infection with influenza B virus (B/Massachusetts/2/2012). Genes associated with viral infection and immune responses against viruses are shown in orange dots. Significantly up-regulated and down-regulated gene profiling are represented as red and blue dots, respectively. (B) Gene Ontology (GO) enrichment analysis performed based on up-regulated and down-regulated gene lists of IRF7−/− MDCK compared to WT MDCK.

Influenza whole genome sequencing

To test whether the virus propagated from IRF7−/− MDCK might be mutated, the whole genome sequencing of influenza B virus (B/Massachusetts/2/2012) was compared between the viral progeny obtained from WT and IRF7−/− MDCK cells. The sequencing data yielded more than 0.5 million reads with an average genome coverage of more than 3,700X for both virus samples. The summary of influenza whole-genome sequencing results is shown in Table S1. Pairwise alignment of each consensus segmented gene between virus propagated from WT and IRF7−/− MDCK cells revealed no mutation in the 8 segmented genomes, implying that the viral progenies from IRF7−/− MDCK cells were identical to those found in WT MDCK cells.

IRF7−/− MDCK cells enhanced influenza viral production

To determine the efficiency of IRF7−/− MDCK cells for influenza viral propagation, the RT-qPCR assay and hemagglutination assay (HA) were performed. For the initial investigation, influenza B/Yamagata (B/Massachusetts/2/2012) was used as a seed virus to propagate within the cells for 48 h, and then the viral titers were compared between IRF7−/− and WT MDCK cells. The result has shown that the influenza B/Yamagata viral titers produced from IRF7−/− MDCK cells (128 HA units) were approximately 2-fold significantly increased (p = 1.50e−05) compared to WT MDCK cells (64 HA units) (Figs. 4A and 4C). Then, the investigation was expanded to test for influenza A/pH1N1 (A/Thailand/104/2009). The yields of influenza A/pH1N1 (5-fold increased; p = 4.81e−06) observed in IRF7−/− MDCK indicated significantly higher viral titers when compared to WT MDCK cells supernatant (Fig. 4B). Moreover, the yields of influenza A/pH1N1 viral titers produced from IRF7−/− MDCK cells (16 HA units) were 4-fold significantly increased (p = 1.0e−04) compared to WT MDCK cells (4 HA units) (Fig. 4D). This finding implies that the lower response of the IFN-related gene in the IRF7−/− MDCK cell was resulting in higher yields of influenza viral production.

Figure 4 Quantitation of influenza viral titers in WT MDCK and IRF7−/− MDCK cell line.

(A) The quantity of influenza viral RNA in supernatant of cells infected with B/Yamagata and (B) A/pH1N1 (48 hpi) in IRF7 −/− MDCK compared to WT MDCK. (C) The hemagglutination unit of influenza virus in supernatant propagated from cells infected with B/Yamagata and (D) A/pH1N1 (48 hpi) in IRF7−/− MDCK compared to WT MDCK. (*); p-value ≤ 0.05, (***); p-value ≤ 0.001.

Discussion

Influenza viruses still cause major problems in global health. The best way to prevent these viruses is through vaccination. However, the vaccine production in embryonated eggs is limited by the constraints of being time-consuming, having a high cost, and contaminating by egg protein (Manini et al., 2017). A cell-based strategy is faster and cheaper for vaccine production than an egg-based one. Still, the host immune response of mammalian cells might limit viral propagation in the cells, resulting in lower viral titers than embryonated eggs. Recently, a study revealed a way to produce an influenza vaccine with higher viral titers by diminishing the host immune response. The previous study investigated the effect of 23 target genes being silenced by shRNA for viral production. The result showed that the MDCK cell knocked down by IRF7 shRNA leads to enhanced influenza propagation (Hamamoto et al., 2013). The percentage expression of IRF7 in knocked-down MDCK cells was approximately 30% compared to WT MDCK cells in an uninfected condition. Based on the IRF7 knocked-out in our study, the relative expression of IRF7 was significantly lower in IRF7−/− MDCK cells compared to WT MDCK cells under infected (95.22% RNA & 32.27% protein reduction) and uninfected (94.76% RNA & 54.85% protein reduction) conditions (Figs. 2A and 2B). Obviously, the IRF7 knock-out strategy based on CRISPR-Cas9 revealed a higher silencing efficiency than the knock-down based on the RNA interference (RNAi) process due to the CRISPR-Cas technique disrupts the gene function in the genome. In contrast, RNAi interferes with the expression level of the mRNA. Moreover, RNAi often exhibits significant off-target effects and unpredictable knock-down efficiencies. On the other hands, CRISPR appears advantageous since knock-down efficiencies appear superior to RNAi (Boettcher & McManus, 2015).

It is not only IRF7 that responds to viral infection, as this is also common to a group of interferon-stimulated genes (ISG), such as Orthomyxovirus resistance gene (Mx), oligoadenylate synthetase (OAS), ISG15, and Interferon-induced transmembrane protein 2 (IFITMs) (Garcia-Sastre, 2011). These interferon stimulating genes and another 41 IFN-related genes were selected to determine the interferon response (Schoggins & Rice, 2011; Schneider, Chevillotte & Rice, 2014). Our result showed lower expression of these genes in infected IRF7−/− than WT MDCK cells. It might be an indication that the interferon responses in this cell were diminished by IRF7 disruption (Fig. 2C). Moreover, Gene Ontology enrichment analysis in IRF7−/− MDCK showed down-regulation of the defense response against the virus as well as IFN-gamma production (Fig. 3B). However, the up-regulation of NF-kB and innate immune response in IRF7−/− MDCK might be due to the gene editing process and the compensations of anti-viral mechanism. NF-kB generally responds to stimuli and triggers cell apoptosis (Baichwal & Baeuerle, 1997). In a previous study, the viral production of knocked-down MDCK cells yielded 4-fold and 5-fold increases in influenza A/H1N1(PR8) and A/pH1N1 viral titers, respectively (Hamamoto et al., 2013). Similar to the previous study, influenza (A/pH1N1) viral titers were enhanced by approximately 4 to 5-fold increases in IRF7−/− MDCK cells knocked out by CRISPR-Cas9 (Figs. 4B and 4D). In addition, the viral titers of influenza B/Yamagata (approximately 2-fold increasing) were found in IRF7−/− MDCK cells (Figs. 4A and 4C). In Fig. 4, the effects of IRF7 knock-out on viral titers depend on viral strains because the NS1 protein of influenza A virus counters host anti-viral defenses by antagonizing the type I interferon (IFN) response (Tisoncik et al., 2011). In contrast, the NS1 protein of the influenza B virus antagonizes the beta interferon induction (Dauber, Heins & Wolff, 2004). Therefore, the impacts of IRF7 knock-out on the influenza A and B viruses were different.

Although the IRF7 was the attractive candidate gene for high viral titers, there were also other target genes for increasing influenza virus production. The increased influenza A viral production in MDCK and VERO cells with Bone Marrow Stromal Cell Antigen 2 (BST - 2) gene knocked out based on the TALENT technique has been previously reported (Yi et al., 2017). There was an approximately 2 to 5-fold increase of influenza A viral titers in supernatant from BST-2 deficient MDCK cells and 6 to 50-fold from BST-2 deficient VERO cells (Yi et al., 2017). Nevertheless, the different fold increasing of influenza virus in various reports could be affected by several factors such as cell types, viral strains, infectious dose, incubation time, and status of cells.

In conclusion, the IRF7−/− MDCK cells provided higher viral titers of influenza viruses which would be attractive for greater capacity and efficiency of influenza vaccine production. However, the IRF7 knock-out based on CRISPR-Cas9 in our study has a limitation in terms of using the polyclonal cell lines. There were mixed populations of MDCK cells (IRF7−/−, IRF7+/− and IRF7+/+) due to approximately 60% gene modification efficiency (Fig. 1C). Thus, the expression of IRF7 protein was still observed in the polyclonal cell lines (Fig. 2B). Ideally, monoclonal knock-out cell lines should be propagated from a single knock-out cell clone to ensure that all cells are the same genotype (homozygous knock-out). The IRF7 protein expression would be completely diminished in the monoclonal knock-out cells, yielding even higher influenza virus production. Therefore, the monoclonal knock-out (IRF7−/− MDCK) cells based on the CRISPR-Cas strategy would be attractive for further improvement of influenza vaccine production.

Supplemental Information

Supplemental Information 1 Representative flow cytometry plots for the sorting of OFP-expressing MDCK cells to potentially select IRF7 knock-out cells based on CRISPR-Cas9

(A) and (B) represent negative control MDCK cells. (C) and (D) represent IRF7−/− MDCK cells. Cells with high OFP expression levels (P2 area) were isolated by fluorescence-activated cell sorting (FACS).

Click here for additional data file.

Supplemental Information 2 Summary result of influenza B (B/Massachusetts/2/2012) whole genome sequencing analysis

Click here for additional data file.

Supplemental Information 3 Raw data for RT-qPCR and the quantity of influenza viruses (Figs. 2 and 4)

The Ct values obtained from RT-qPCR were used for the calculation of IRF7 gene expression and influenza viral titers. The results were illustrated as Figs. 2 and 4, respectively.

Click here for additional data file.

Supplemental Information 4 Raw image Fig. 1

Raw image of agarose gel electrophoresis of cleavage detection displayed the parental band and cleavage bands from IRF7 −/− MDCK.

Click here for additional data file.

Supplemental Information 5 The relative expression of IRF7 protein based on western blot in IRF7−/− MDCK and WT MDCK during uninfected and infected with influenza B virus (12 hpi)

Click here for additional data file.

We would like to express our gratefulness to Assoc. Prof. Nipan Israsena, Ph.D. and Praewphan Ingrungruanglert, Ph.D. (Center of Excellence for Stem Cell and Cell Therapy, Faculty of Medicine, Chulalongkorn University) for cell sorting. We also would like to thank Nuttiya Kalpongnukul, Ph.D. (Research Affairs, Faculty of Medicine, Chulalongkorn University) for western blot analysis.

Additional Information and Declarations

Competing Interests

Author Contributions

DNA Deposition

Data Availability

The authors declare there are no competing interests.

Oraphan Mayuramart performed the experiments, analyzed the data, prepared figures and/or tables, authored or reviewed drafts of the article, and approved the final draft.

Witthaya Poomipak conceived and designed the experiments, performed the experiments, authored or reviewed drafts of the article, and approved the final draft.

Somruthai Rattanaburi performed the experiments, authored or reviewed drafts of the article, and approved the final draft.

Kritsada Khongnomnan performed the experiments, authored or reviewed drafts of the article, and approved the final draft.

Songtham Anuntakarun analyzed the data, authored or reviewed drafts of the article, and approved the final draft.

Suthat Saengchoowong performed the experiments, authored or reviewed drafts of the article, and approved the final draft.

Tanit Chavalit analyzed the data, authored or reviewed drafts of the article, and approved the final draft.

Naphat Chantaravisoot analyzed the data, prepared figures and/or tables, authored or reviewed drafts of the article, and approved the final draft.

Sunchai Payungporn conceived and designed the experiments, analyzed the data, authored or reviewed drafts of the article, and approved the final draft.

The following information was supplied regarding the deposition of DNA sequences:

The data is available at GenBank: PRJNA827931.

Link: https://www.ncbi.nlm.nih.gov/sra/PRJNA827931.

Accession: PRJNA827931.

The following information was supplied regarding data availability:

Data is available at GenBank: PRJNA827931.

Link: https://www.ncbi.nlm.nih.gov/sra/PRJNA827931

The data were released on 15 August 2022.

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
