# Peer review of "IRF7-deficient MDCK cell based on CRISPR/Cas9 technology for enhancing influenza virus replication and improving vaccine production"

_PeerJ, doi:10.7717/peerj.13989_

## Round 0.1 · original submission · Major Revisions

Please answer the questions from reviewers, and explain their concerns, and address the issues reviewers pointed out.

Reviewer 1 ·

Basic reporting

Mayuramart et al have reported that knock-down of the IRF7 gene in MDCK can improve the titer of influenza viruses. They provided clear rationales/goals for this study. The study may be helpful in improving the strategy to make influenza vaccination efficient. I do not have any issues with this manuscript except for the following points.

1. Fig.1C is too dark to see the cleaved products, particularly in the printed manuscript. I would suggest showing the enhanced contrast figures.

2. The authors examined transcriptional levels of IRF7 and other genes. At least, I would suggest examining the protein levels of IRF7.

3. Fig. 2A-B, the authors showed the values of uninfected and infected, separately. I would suggest showing relative values to the uninfected WT because it will allow us to see how the viral infection caused the upregulation of IRF7 with CRISPR.

Experimental design

The authors provided sufficient details of the materials and methods used in this manuscript.
They offered clear rationales/goals for this study.

I would suggest emphasizing the advantage of CRISPR techniques over the previous shRNA method in the discussion section.

Validity of the findings

Fig. 4 shows the effects of IRF7 KO on viral titers using several viral strains, but it looks depending on viral strains. Is there any information about the difference in counterpart proteins against IFN among viral strains? I would also suggest mentioning the limitation of this study in the discussion part because the impacts of B strains are not so large.

Reviewer 2 ·

Basic reporting

Oraphan Mayuramartet, al. has developed IRF7-/- MDCK which can be used for production of high quantity of Influenza viruses for the production of vaccines. Authors describe the process of generating IRF7-/- MDCK cell line by CRISPR Cas9 technology, which is much more efficient in generating knockout cell line than the previously used shRNA knockdown. Influenza viruses replicate more efficiently in the IRF7-/- MDCK cell as compared to WT due to downregulation of various ISGs downstream of IRF7 pathway. Authors also sequence and confirm that there are no mutations in the viruses generated through replication in the IRF7-/- MDCK cell line and thus can be safely used for vaccine production.
Major points
• Its not clear in the methods section whether IRF7-/- MDCK cell line was generated from single cell after sorting or it’s a polyclonal cell line.
• Quantitative PCR data has been used to show the relative expression of IRF7 in WT and IRF7-/- MDCK cell line. Relative expression of IRF7 is reduced by 94.76% in IRF7-/- MDCK cells. Western blot data confirming absence of IRF7 protein is needed to say that it’s a knockout cell line. For generating CRISPR knockout cell line its recommended to collect many single cell knockout clones. Ideally it would have been nice to have cell line from at least two clones to check the effect on Influenza virus replication. And choose the clone which show the maximum effect and that cell line can then be deployed for vaccine production.

Minor points
• Line 31- Rephrase the sentence: The expression of IRF7 in IRF7-/- MDCK was reduced by 94.76% in uninfected and 82.14% under viral infected conditions.
• Line 55, 56- Use the word two instead of number 2.
• Line 69- Rephrase the sentence: Type I Interferon (IFN) is an important mediator of innate immune system that responds to viral infection.
• Line 92- Correct the sentence; You have not developed interferon defective MDCK cell line.
• Line 164- Grammar: WT and IRF7-/- MDCK cells were seeded onto T25 cell culture flask.

Experimental design

Experiments are well designed and explained.

Validity of the findings

Findings are well described.

Reviewer 3 ·

Basic reporting

The manuscript submitted by Oraphan Mayuramart et al. and the team has done a very interesting study on IRF7 deficient MDCK cells for the influenza virus replication and the IRF-7 knockout cells might be useful for the vaccine production. The manuscript is well written, however needs major revision.

Experimental design

Major revisions-

1. MDCK cells were used for the transfection, what was the transfection efficiency of the transfected cells, because the MDCK cells are hard to transfect cell lines. Explain?
2. The author has written that after transfection the OFP positive cells were sorted for further downstream assays, what was OFP positive cell population in the transfected cells, Please add the gated positive cells in the supplementary figure.
3. The virus titer should be checked after influenza (strains used) virus infection between WT MDCK and IRF7 -/- MDCK cells.
4. In fig3, there are A, B and C figures, but in figure legends there is a lack of detail in the graph in figure legends, correct the figure legend.
5. In table1, it has been mentioned that IRF-7_F2060-TS and IRF-7_F2060-BS; are this different gRNA? Ideally, for the IRF-7 gene (used in the study) three gRNA should be taken for the knockout study.
6. In fig4, three graphs depicting the virus copy no, In the results, it has been written that two virus-specific genes (M and PB1) have been taken but there are only three graphs in the figure, please explain.
7. In discussion, lines 318-320, it is not clear what was the basis of IRF-7 expression calculation in FRT PCR, relative or absolute quantification, please clarify.
8. In results lines 318-320, it has been written that there were various IFNs genes were dysregulated (down-regulated), apart from this write the total no of dysregulated genes between WT MDCK and IRF7-/- MDCK cells infected with influenza virus. The graph should be there of differentially expressed genes. Please explain what is the difference between the differential expression in the groups (WT and IRF-7-/- KO MDCK cells).

Validity of the findings

No comments

Additional comments

Minor revision-
1. In fig 4, write A,B,C on the graphs, because in the result section it was written that fig 4A, 4B, and 4c, but there was no A,B and C has been written on fig 4.
2. In Fig 1, highlight the sgRNA sequence region and PAM region.

---

## Round 0.2 · Minor Revisions

The reviewer wants authors to mention this in manuscript clearly that ideal approach is to use monoclonal knockout cell line which would give clean knockout cell line and may produces higher Influenza virus titer, please mention the limitation in manuscript.

Reviewer 2 ·

Basic reporting

No comments

Experimental design

Ideal condition for

Validity of the findings

After first revision, authors have provided the western blot data showing the expression of IRF7 protein. It seems that there is more than enough IRF7 protein left. Because authors started from polyclonal cells there is a mix population. I would want authors to mention this in manuscript clearly that ideal approach would have been to use monoclonal knockout cell line which would give clean knockout cell line and have given even higher Influenza virus titer.

---

## Round 0.3 · accepted · Accept

I am pleased to inform you that your revised manuscript has been accepted now since you addressed the requirement the reviewer mentioned, congrats!